# Intrinsic Motivation and Automatic Curricula via Asymmetric Self-Play

**Sainbayar Sukhbaatar**
Dept. of Computer Science
New York University
sainbar@cs.nyu.edu

**Zeming Lin**
Facebook AI Research
New York
zlin@fb.com

**Ilya Kostrikov**
Dept. of Computer Science
New York University
kostrikov@cs.nyu.edu

**Gabriel Synnaeve, Arthur Szlam & Rob Fergus**
Facebook AI Research
New York
{gab,aszlam,robfergus}@fb.com

## Abstract

We describe a simple scheme that allows an agent to learn about its environment in an unsupervised manner. Our scheme pits two versions of the same agent, Alice and Bob, against one another. Alice proposes a task for Bob to complete; and then Bob attempts to complete the task. In this work we will focus on two kinds of environments: (nearly) reversible environments and environments that can be reset. Alice will "propose" the task by doing a sequence of actions and then Bob must undo or repeat them, respectively. Via an appropriate reward structure, Alice and Bob automatically generate a curriculum of exploration, enabling unsupervised training of the agent. When Bob is deployed on an RL task within the environment, this unsupervised training reduces the number of supervised episodes needed to learn, and in some cases converges to a higher reward.

## 1 Introduction

Model-free approaches to reinforcement learning are sample inefficient, typically requiring a huge number of episodes to learn a satisfactory policy. The lack of an explicit environment model means the agent must learn the rules of the environment from scratch at the same time as it tries to understand which trajectories lead to rewards. In environments where reward is sparse, only a small fraction of the agents' experience is directly used to update the policy, contributing to the inefficiency.

In this paper we introduce a novel form of unsupervised training for an agent that enables exploration and learning about the environment without any external reward that incentivizes the agents to learn how to transition between states as efficiently as possible. We demonstrate that this unsupervised training allows the agent to learn new tasks within the environment quickly.

## 2 Approach

We consider environments with a single physical agent (or multiple physical units controlled by a single agent), but we allow it to have two separate "minds": Alice and Bob, each with its own objective and parameters. During self-play episodes, Alice's job is to propose a task for Bob to complete, and Bob's job is to complete the task. When presented with a target task episode, Bob is then used to perform it (Alice plays no role). The key idea is that the Bob's play with Alice should help him understand how the environment works and enabling him to learn the target task more quickly.

Our approach is restricted to two classes of environment: (i) those that are (nearly) reversible, or (ii) ones that can be reset to their initial state (at least once). These restrictions allow us to sidestep complications around how to communicate the task and determine its difficulty (see Appendix F.2

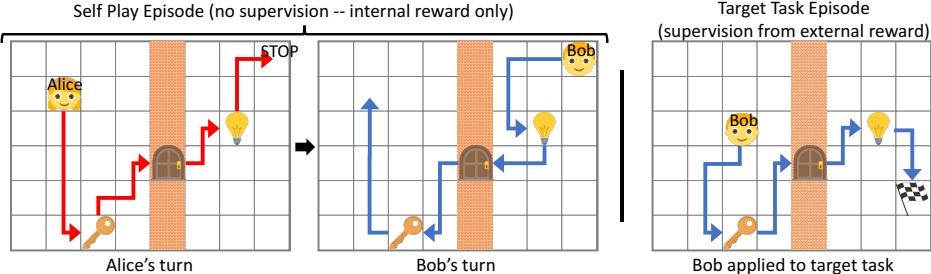

Figure 1: Illustration of the self-play concept in a gridworld setting. Training consists of two types of episode: self-play and target task. In the former, Alice and Bob take turns moving the agent within the environment. Alice sets tasks by altering the state via interaction with its objects (key, door, light) and then hands control over to Bob. He must return the environment to its original state to receive an internal reward. This task is just one of many devised by Alice, who automatically builds a curriculum of increasingly challenging tasks. In the target task, Bob's policy is used to control the agent, with him receiving an *external* reward if he visits the flag. He is able to learn to do this quickly as he is already familiar with the environment from self-play.

for further discussion). In these two scenarios, Alice starts at some initial state $s_0$ and proposes a task by *doing* it, i.e. executing a sequence of actions that takes the agent to a state $s_t$. She then outputs a STOP action, which hands control over to Bob. In reversible environments, Bob's goal is to return the agent back to state $s_0$ (or within some margin of it, if the state is continuous), to receive reward. In partially observable environments, the objective is relaxed to Bob finding a state that returns the same observation as Alice's initial state. In environments where resets are permissible, Alice's STOP action also reinitializes the environment, thus Bob starts at $s_0$ and now must reach $s_t$ to be rewarded, thus repeating Alice's task instead of reversing it. See Fig. 1 for an example, and also Algorithm 1 in Appendix A.

In both cases, this self-play between Alice and Bob only involves internal reward (detailed below), thus the agent can be trained without needing any supervisory signal from the environment. As such, it comprises a form of unsupervised training where Alice and Bob explore the environment and learn how it operates. This exploration can be leveraged for some target task by training Bob on target task episodes in parallel. The idea is that Bob's experience from self-play will help him learn the target task in fewer episodes. The reason behind choosing Bob for the target task is because he learns to transfer from one state to another efficiently from self-play. See Algorithm 2 in Appendix A for detail.

For self-play, we choose the reward structure for Alice and Bob to encourage Alice to push Bob past his comfort zone, but not give him impossible tasks. Denoting Bob's total reward by $R_B$ (given at the end of episodes) and Alice's total reward by $R_A$, we use

$$R_B = -\gamma t_B \tag{1}$$

where $t_B$ is the time taken by Bob to complete his task and

$$R_A = \gamma \max(0, t_B - t_A) \tag{2}$$

where $t_A$ is the time until Alice performs the STOP action, and $\gamma$ is a scaling coefficient that balances this internal reward to be of the same scale as external rewards from the target task. The total length of an episode is limited to $t_{\text{Max}}$, so if Bob fails to complete the task in time we set $t_B = t_{\text{Max}} - t_A$.

Thus Alice is rewarded if Bob takes more time, but the negative term on her own time will encourage Alice not to take too many steps when Bob is failing. For both reversible and resettable environments, Alice must limit her steps to make Bob's task easier, thus Alice's optimal behavior is to the find simplest tasks that Bob cannot complete. This eases learning for Bob since the new task will be only just beyond his current capabilities. The self-regulating feedback between Alice and Bob allows them to automatically construct a curriculum for exploration, a key contribution of our approach.

## 2.1 Parameterizing Alice and Bob's actions

Alice and Bob each have policy functions which take as input two observations of state variables, and output a distribution over actions. In Alice's case, the function will be of the form

$$a_A = \pi_A(s_t, s_0),$$

where $s_0$ is the observation of the initial state of the environment and $s_t$ is the observation of the current state. In Bob's case, the function will be

$$a_B = \pi_B(s_t, s^*),$$

where $s^*$ is the target state that Bob has to reach, and set to $s_0$ when we have a reversible environment. In a resettable environment $s^*$ is the state where Alice executed the STOP action.

When a target task is presented, the agent's policy function is $a_{\text{Target}} = \pi_B(s_t, \emptyset)$, where the second argument of Bob's policy is simply set to zero [1]. If $s^*$ is always non-zero, then this is enough to let Bob know whether the current episode is self-play or target task. In some experiments where $s^*$ can be zero, we give third argument $z \in \{0, 1\}$ that explicitly indicates the episode kind.

In the experiments below, we demonstrate our approach in settings where $\pi_A$ and $\pi_B$ are tabular; where it is a neural network taking discrete inputs, and where it is a neural network taking in continuous inputs. When using a neural network, we use the same network architecture for both Alice and Bob, except they have different parameters

$$\pi_A(s_t, s_0) = f(s_t, s_0, \theta_A), \quad \pi_B(s_t, s^*) = f(s_t, s^*, \theta_B),$$

where $f$ is an multi-layered neural network with parameters $\theta_A$ or $\theta_B$.

## 2.2 Universal Bob in the tabular setting

We now present a theoretical argument that shows for environments with finite states, tabular policies, and deterministic, Markovian transitions, we can interpret the self-play as training Bob to find a policy that can get from any state to any other in the least expected number of steps.

**Preliminaries:** Note that, as discussed above, the policy table for Bob is indexed by $(s_t, s^*)$, not just by $s_t$. In particular, with the assumptions above, this means that there is a *fast policy* $\pi_{\text{fast}}$ such that $\pi_{\text{fast}}(s_t, s^*)$ has the smallest expected number of steps to transition from $s_t$ to $s^*$. It is clear that $\pi_{\text{fast}}$ is a universal policy for Bob, such that $\pi_B = \pi_{\text{fast}}$ is optimal with respect to any Alice's policy $\pi_A$. In a reset game, $\pi_{\text{fast}}$ nets Alice a return of 0, and in the reverse game, the return of $\pi_{\text{fast}}$ against an optimal Alice can be considered a measure of the reversibility of the environment. However, in what follows let us assume that either the reset game or the reverse game in a perfectly reversible environment is used. Also, let assume the initial states are randomized and its distribution covers the entire state space.

**Claim:** If $\pi_A$ and $\pi_B$ are policies of Alice and Bob that are in equilibrium (i.e., Alice cannot be made better without changing Bob, and vice-versa), then $\pi_B$ is a fast policy.

**Argument:** Let us first show that Alice will always get zero reward in equilibrium. If Alice is getting positive reward on some challenge, that means Bob is taking longer than Alice on that challenge. Then Bob can be improved to use $\pi_{fast}$ at that challenge, which contradicts the equilibrium assumption.

Now let us prove $\pi_B$ is a fast policy by contradiction. If $\pi_B$ is not fast, then there must exist a challenge $(s_t, s^*)$ where $\pi_B$ will take longer than $\pi_{fast}$. Therefore Bob can get more reward by using $\pi_{fast}$ if Alice does propose that challenge with non-zero probability. Since we assumed equilibrium and $\pi_B$ cannot be improved while $\pi_A$ fixed, the only possibility is that Alice is never proposing that challenge. If that is true, Alice can get positive reward by proposing that task using the same actions as $\pi_{fast}$, so taking fewer steps than $\pi_B$. However this contradicts with the proof that Alice always gets zero reward, making our initial assumption "$\pi_B$ is not fast" wrong.

---

[1] Note that Bob can be used in multi-task learning by feeding the task description into $\pi_B$

## 3   RELATED WORK

Self-play arises naturally in reinforcement learning, and has been well studied. For example, for playing checkers (Samuel, 1959), backgammon (Tesauro, 1995), and Go, (Silver et al., 2016), and in multi-agent games such as RoboSoccer (Riedmiller et al., 2009). Here, the agents or teams of agents compete for external reward. This differs from our scheme where the reward is purely internal and the self-play is a way of motivating an agent to learn about its environment to augment sparse rewards from separate target tasks.

Our approach has some relationships with generative adversarial networks (GANs) (Goodfellow et al., 2014), which train a generative neural net by having it try to fool a discriminator network which tries to differentiate samples from the training examples. Li et al. (2017) introduce an adversarial approach to dialogue generation, where a generator model is subjected to a form of "Turing test" by a discriminator network. Mescheder et al. (2017) demonstrate how adversarial loss terms can be combined with variational auto-encoders to permit more accurate density modeling. While GAN's are often thought of as methods for training a generator, the generator can be thought of as a method for generating hard negatives for the discriminator. From this viewpoint, in our approach, Alice acts as a "generator", finding "negatives" for Bob. However, Bob's jobs is to complete the generated challenge, not to discriminate it.

There is a large body of work on intrinsic motivation (Barto, 2013; Singh et al., 2004; Klyubin et al., 2005; Schmidhuber, 1991) for self-supervised learning agents. These works propose methods for training an agent to explore and become proficient at manipulating its environment without necessarily having a specific target task, and without a source of extrinsic supervision. One line in this direction is curiosity-driven exploration (Schmidhuber, 1991). These techniques can be applied in encouraging exploration in the context of reinforcement learning, for example (Bellemare et al., 2016; Strehl & Littman, 2008; Lopes et al., 2012; Tang et al., 2016; Pathak et al., 2017); Roughly, these use some notion of the novelty of a state to give a reward. In the simplest setting, novelty can be just the number of times a state has been visited; in more complex scenarios, the agent can build a model of the world, and the novelty is the difficulty in placing the current state into the model. In our work, there is no explicit notion of novelty. Even if Bob has seen a state many times, if he has trouble getting to it, Alice should force him towards that state. Another line of work on intrinsic motivation is a formalization of the notion of empowerment (Klyubin et al., 2005), or how much control the agent has over its environment. Our work is related in the sense that it is in both Alice's and Bob's interests to have more control over the environment; but we do not explicitly measure that control except in relation to the tasks that Alice sets.

Curriculum learning (Bengio et al., 2009) is widely used in many machine learning approaches. Typically however, the curriculum requires at least some manual specification. A key point about our work is that Alice and Bob devise their own curriculum entirely automatically. Previous automatic approaches, such as Kumar et al. (2010), rely on monitoring training error. But since ours is unsupervised, no training labels are required either.

Our basic paradigm of "Alice proposing a task, and Bob doing it" is related to the Horde architecture (Sutton et al., 2011) and (Schaul et al., 2015). In those works, instead of using a value function $V = V(s)$ that depends on the current state, a value function that explicitly depends on state and goal $V = V(s, g)$ is used. In our experiments, our models will be parameterized in a similar fashion. The novelty in this work is in how Alice defines the goal for Bob.

The closest work to ours is that of Baranes & Oudeyer (2013), who also have one part of the model that proposes tasks, while another part learns to complete them. As in this work, the policies and cost are parameterized as functions of both state and goal. However, our approach differs in the way tasks are proposed and communicated. In particular, in Baranes & Oudeyer (2013), the goal space has to be presented in a way that allows explicit partitioning and sampling, whereas in our work, the goals are sampled through Alice's actions. On the other hand, we pay for not having to have such a representation by requiring the environment to be either reversible or resettable.

Several concurrent works are related: Andrychowicz et al. (2017) form an implicit curriculum by using internal states as a target. Florensa et al. (2017) automatically generate a series of increasingly distant start states from a goal. Pinto et al. (2017) use an adversarial framework to perturb the

environment, inducing improved robustness of the agent. Held et al. (2017) propose a scheme related to our "random Alice" strategy[2].

## 4 EXPERIMENTS

The following experiments explore our self-play approach on a variety of tasks, both continuous and discrete, from the Mazebase (Sukhbaatar et al., 2015), RLLab (Duan et al., 2016), and Star-Craft (Synnaeve et al., 2016) environments. The same protocol is used in all settings: self-play and target task episodes are mixed together and used to train the agent via discrete policy gradient. We evaluate both the reverse and repeat versions of self-play. We demonstrate that the self-play episodes help training, in terms of number of target task episodes needed to learn the task. Note that we assume the self-play episodes to be "free", since they make no use of environmental reward. This is consistent with traditional semi-supervised learning, where evaluations typically are based only on the number of labeled points (not unlabeled ones too).

In all the experiments we use policy gradient (Williams, 1992) with a baseline for optimizing the policies. In the tabular task below, we use a constant baseline; in all the other tasks we use a policy parameterized by a neural network, and a baseline that depends on the state. We denote the states in an episode by $s_1, ..., s_T$, and the actions taken at each of those states as $a_1, ..., a_T$, where $T$ is the length of the episode. The baseline is a scalar function of the states $b(s, \theta)$, computed via an extra head on the network producing the action probabilities. Besides maximizing the expected reward with policy gradient, the models are also trained to minimize the distance between the baseline value and actual reward. Thus after finishing an episode, we update the model parameters $\theta$ by

$$\Delta\theta = \sum_{t=1}^{T} \left[ \frac{\partial \log f(a_t|s_t, \theta)}{\partial \theta} \left( \sum_{i=t}^{T} r_i - b(s_t, \theta) \right) - \lambda \frac{\partial}{\partial \theta} \left( \sum_{i=t}^{T} r_i - b(s_t, \theta) \right)^2 \right].$$

Here $r_t$ is reward given at time $t$, and the hyperparameter $\lambda$ is for balancing the reward and the baseline objectives, which is set to 0.1 in all experiments.

For the policy neural networks, we use two-layer fully-connected networks with 50 hidden units in each layer. The training uses RMSProp (Tieleman & Hinton, 2012). We always do 10 runs with different random initializations and report their mean and standard deviation. See Appendix B for all the hyperparameter values used in the experiments.

### 4.1 LONG HALLWAY

We first describe a simple toy environment designed to illustrate the function of the asymmetric self-play. The environment consists of $M$ states $\{s_1, ..., s_M\}$ arranged in a chain. Both Alice and Bob have three possible actions, "left", "right", or "stop". If the agent is at $s_i$ with $i \neq 1$, "left" takes it to $s_{i-1}$; "right" analogously increases the state index, and "stop" transfers control to Bob when Alice runs it and terminates the episode when Bob runs it. We use "return to initial state" as the self-play task (i.e. Reverse in Algorithm 1 in Appendix A ). For the target task, we randomly pick a starting state and target state, and the episode is considered successful if Bob moves to the target state and executes the stop action before a fixed number of maximum steps.

In this case, the target task is essentially the same as the self-play task, and so running it is not unsupervised learning (and in particular, on this toy example unlike the other examples below, we do not mix self-play training with target task training). However, we see that the curriculum afforded by the self-play is efficient at training the agent to do the target task at the beginning of the training, and is effective at forcing exploration of the state space as Bob gets more competent.

In Fig. 2 (left) we plot the number of episodes vs rate of success at the target task with four different methods. We set $M = 25$ and the maximum allowed steps for Alice and Bob to be 30. We use fully tabular controllers; the table is of size $M^2 \times 3$, with a distribution over the three actions for each possible (start, end pair).

The red curve corresponds to policy gradient, with a penalty of $-1$ given upon failure to complete the task, and a penalty of $-t/t_{\text{Max}}$ for successfully completing the task in $t$ steps. The magenta curve

---

[2]In their paper they analyzed our approach, suggesting it was inherently unstable. However, the analysis relied on a sudden jump of Bob policy with respect to Alice's, which is unlikely to happen in practice.

corresponds to taking Alice to have a random policy (1/2 probability of moving left or right, and not stopping till the maximum allowed steps). The green curve corresponds to policy gradient with an exploration bonus similar to Strehl & Littman (2008). That is, we keep count of the number of times $N_s$ the agent has been in each state $s$, and the reward for $s$ is adjusted by exploration bonus $\alpha/\sqrt{N_s}$, where $\alpha$ is a constant balancing the reward from completing the task with the exploration bonus. We choose the weight $\alpha$ to maximize success at 0.2M episodes from the set $\{0, 0.1, 0.2, ..., 1\}$. The blue curve corresponds to the asymmetric self-play training.

We can see that at the very beginning, a random policy for Alice gives some form of curriculum but eventually is harmful, because Bob never gets to see any long treks. On the other hand, policy gradient sees very few successes in the beginning, and so trains slowly. Using the self-play method, Alice gives Bob easy problems at first (she starts from random), and then builds harder and harder problems as the training progresses, finally matching the performance boost of the count based exploration. Although not shown, similar patterns are observed for a wide range of learning rates.

## 4.2 MAZEBASE: LIGHT KEY

We now describe experiments using the MazeBase environment (Sukhbaatar et al., 2015), which have discrete actions and states, but sufficient combinatorial complexity that tabular methods cannot be used. The environment consist of various items placed on a finite 2D grid; and randomly generated for each episode.

We use an environment where the maze contains a light switch (whose initial state is sampled according to a predefined probability, p(Light off)), a key and a wall with a door (see Fig. 1). An agent can open or close the door by toggling the key switch, and turn on or off light with the light switch. When the light is off, the agent can only see the (glowing) light switch. In the target task, there is also a goal flag item, and the objective of the game is reach to that goal flag.

In self-play, the environment is the same except there is no specific objective. An episode starts with Alice in control, who can navigate through the maze and change the switch states until she outputs the STOP action. Then, Bob takes control and tries to return everything to its original state (restricted to visible items) in the reverse self-play. In the repeat version, the maze resets back to its initial state when Bob takes the control, who tries to reach the final state of Alice.

In Fig. 2 (right), we set p(Light off)=0.5 during self-play[3] and evaluate the repeat form of self-play, alongside two baselines: (i) target task only training (i.e. no self-play) and (ii) self-play with a random policy for Alice. With self-play, the agent succeeds quickly while target task-only training takes much longer[4]. Fig. 3 shows details of a single training run, demonstrating how Alice and Bob automatically build a curriculum between themselves though self-play.

## 4.3 RLLAB: MOUNTAIN CAR

We applied our approach to the Mountain Car task in RLLab. Here the agent controls a car trapped in a 1-D valley. It must learn to build momentum by alternately moving to the left and right, climbing higher up the valley walls until it is able to escape. Although the problem is presented as continuous, we discretize the 1-D action space into 5 bins (uniformly sized) enabling us to use discrete policy gradient, as above. We also added a secondary action head with binary actions to be used as STOP action. An observation of state $s_t$ consists of the location and speed of the car.

As in Houthooft et al. (2016); Tang et al. (2016), a reward of +1 is given only when the car succeeds in climbing the hill. In self-play, Bob succeeds if $\|s_b - s_a\| < 0.2$, where $s_a$ and $s_b$ are the final states (location and velocity of the car) of Alice and Bob respectively.

The nature of the environment makes it highly asymmetric from Alice and Bob's point of view, since it is far easier to coast down the hill to the starting point that it is to climb up it. Hence we exclusively use the reset form of self-play. In Fig. 4 (left), we compare this to current state-of-the-art methods, namely VIME (Houthooft et al., 2016) and SimHash (Tang et al., 2016). Our approach

---

[3]Changing p(Light off) adjusts the seperation between the self-play and target tasks. For a systematic evaluation of this, please see Appendix C.1 .

[4]Training was stopped for all methods except target-only at $5 \times 10^6$ episodes.

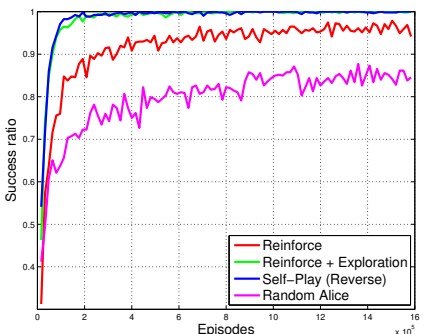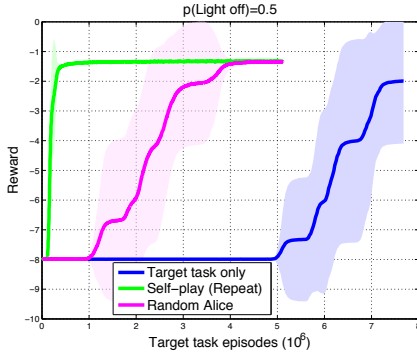

Figure 2: **Left:** The hallway task from section 4.1. The $y$ axis is fraction of successes on the target task, and the $x$ axis is the total number of training examples seen. Standard policy gradient (red) learns slowly. Adding an explicit exploration bonus (Strehl & Littman, 2008) (green) helps significantly. Our self-play approach (blue) gives similar performance however. Using a random policy for Alice (magenta) drastically impairs performance, showing the importance of self-play between Alice and Bob. **Right:** Mazebase task, illustrated in Fig. 1, for p(Light off) = 0.5. Augmenting with the repeat form of self-play enables significantly faster learning than training on the target task alone and random Alice baselines.

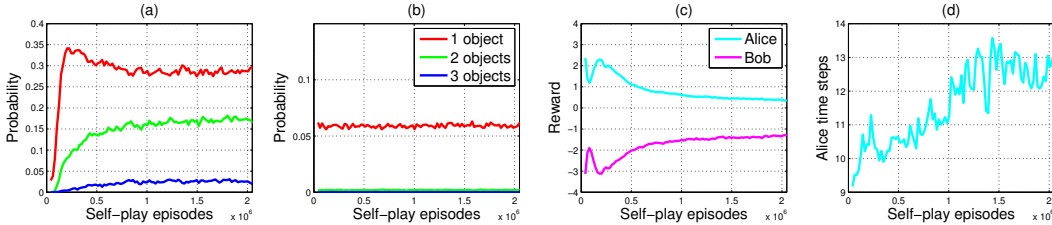

Figure 3: Inspection of a Mazebase learning run, using the environment shown in Fig. 1. (a): rate at which Alice interacts with 1, 2 or 3 objects during an episode, illustrating the automatically generated curriculum. Initially Alice touches no objects, but then starts to interact with one. But this rate drops as Alice devises tasks that involve two and subsequently three objects. (b) by contrast, in the random Alice baseline, she never utilizes more than a single object and even then at a much lower rate. (c) plot of Alice and Bob's reward, which strongly correlates with (a). (d) plot of $t_a$ as self-play progresses. Alice takes an increasing amount of time before handing over to Bob, consistent with tasks of increasing difficulty being set.

(blue) performs comparably to both of these. We also tried using policy gradient directly on the target task samples, but it was unable to solve the problem.

## 4.4 RLLAB: SWIMMERGATHER

We also applied our approach to the SwimmerGather task in RLLab (which uses the Mujoco (Todorov et al., 2012) simulator), where the agent controls a worm with two flexible joints, swimming in a 2D viscous fluid. In the target task, the agent gets reward +1 for eating green apples and -1 for touching red bombs, which are not present during self-play. Thus the self-play task and target tasks are different: in the former, the worm just swims around but in the latter it must learn to swim towards green apples and away from the red bombs.

The observation state consists of a 13-dimensional vector describing location and joint angles of the worm, and a 20 dimensional vector for sensing nearby objects. The worm takes two real values as an action, each controlling one joint. We add a secondary action head to our models to handle the 2nd joint, and a third binary action head for STOP action. As in the mountain car, we discretize the output space (each joint is given 9 uniformly sized bins) to allow the use of discrete policy gradients.

Bob succeeds in a self-play episode when $\|l_b - l_a\| < 0.3$ where $l_a$ and $l_b$ are the final locations of Alice and Bob respectively. Fig. 4 (right) shows the target task reward as a function of training iteration for our approach alongside state-of-the-art exploration methods VIME (Houthooft et al., 2016) and SimHash (Tang et al., 2016). We demonstrate the generality of the self-play approach by applying it to Reinforce and also TRPO (Schulman et al., 2015) (see Appendix D for details). In both cases, it enables them to gain reward significantly earlier than other methods, although both converge to a similar final value to SimHash. A video of our worm performing the test task can be found at `https://goo.gl/Vsd8Js`.

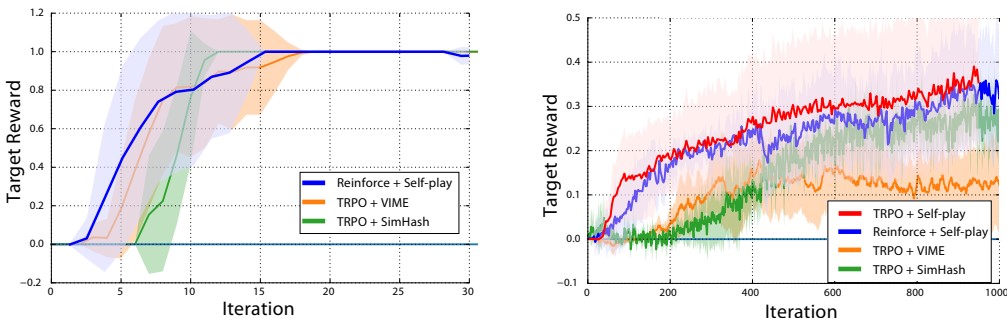

Figure 4: Evaluation on MountainCar (left) and SwimmerGather (right) target tasks, comparing to VIME Houthooft et al. (2016) and SimHash Tang et al. (2016) (figures adapted from Tang et al. (2016)). With reversible self-play we are able to learn faster than the other approaches, although it converges to a comparable reward. Training directly on the target task using Reinforce without self-play resulted in total failure. Here 1 iteration = 5k (50k) target task steps in Mountain car (SwimmerGather), excluding self-play steps.

## 4.5 STARCRAFT: TRAINING MARINES

Finally, we applied our self-play approach to the same setup as the beginning of a standard StarCraft: Brood War game Synnaeve et al. (2016), where an agent controls multiple units to mine, construct buildings, and train new units, but without enemies to fight. The environment starts with 4 workers units (Terran SCVs), who can move around, mine nearby minerals and construct new buildings. In addition, the agent controls the command center, which can train new workers. See Fig. 5 (left) for relations between different units and their actions.

The target task is to build Marine units. To do this, an agent must follow a specific sequence of operations: (i) mine minerals with workers; (ii) having accumulated sufficient mineral supply, build a barracks and (iii) once the barracks are complete, train Marine units out of it. Optionally, an agent can train a new worker for faster mining, or build a supply depot to accommodate more units. When the episode ends after 200 steps (little over 3 minutes), the agent gets rewarded +1 for each Marine it has built. Optimizing this task is highly complex due to several factors. First, the agent has to find an optimal mining pattern (concentrating on a single mineral or mining a far away mineral is inefficient). Then, it has to produce the optimal number of workers and barrack at the right timing. In addition, a supply depot needs to be built when the number of units is close to the limit.

During self-play (repeat variant), Alice and Bob control the workers and can try any combination of actions during the episode. Since exactly matching the game state is almost impossible, Bob's success is only based on the global state of the game, which includes the number of units of each type (including buildings), and accumulated mineral resource. So Bob's objective in self-play is to make as many units and mineral as Alice in shortest possible time. Further details are given in Appendix E. Fig. 5 (right) compares the Reinforce algorithm on the target task, with and without self-play. An additional count-based exploration baseline similar to the hallway experiment is also shown. It utilizes the same global game state as self-play.

## 5 DISCUSSION

In this work we described a novel method for intrinsically motivated learning which we call asymmetric self-play. Despite the method's conceptual simplicity, we have seen that it can be effective in both discrete and continuous input settings with function approximation, for encouraging ex-

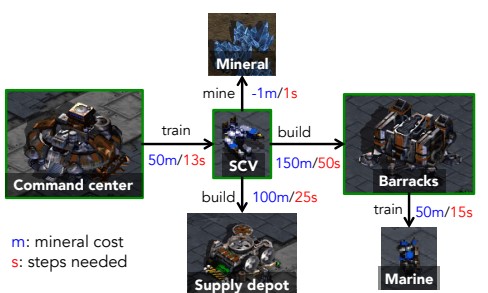 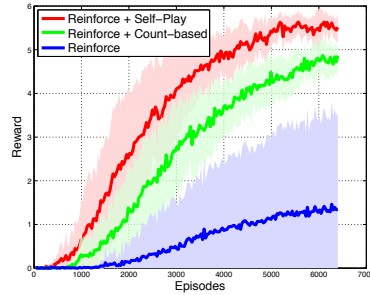

Figure 5: **Left:** Different types of unit in the StarCraft environment. The arrows represent possible actions (excluding movement actions) by the unit, and corresponding numbers shows (blue) amount of minerals and (red) time steps needed to complete. The units under agent's control are outlined by a green border. **Right:** Plot of reward on the StarCraft sub-task of training marine units vs #target-task episodes (self-play episodes are not included), with and without self-play. A count-based baseline is also shown. Self-play greatly speeds up learning, and also surpasses the count-based approach at convergence.

ploration and automatically generating curriculums. On the challenging benchmarks we consider, our approach is at least as good as state-of-the-art RL methods that incorporate an incentive for exploration, despite being based on very different principles. Furthermore, it is possible show theoretically that in simple environments, using asymmetric self-play with reward functions from (1) and (2), optimal agents can transit between any pair of reachable states as efficiently as possible. Code for our approach can be found at (link removed for anonymity).

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

## A    PSEUDO CODE

Algorithm 1 and 2 are the pseudo codes for training an agent on self-play and target task episodes.

---

**Algorithm 1** Pseudo code for training an agent on a self-play episode

---

**function** SELFPLAYEPISODE(REVERSE/REPEAT,$t_{\text{MAX}}, \theta_A, \theta_B$)
    $t_A \leftarrow 0$
    $s_0 \leftarrow$ env.observe()
    $s^* \leftarrow s_0$
    **while** True **do**
        # Alice's turn
        $t_A \leftarrow t_A + 1$
        $s \leftarrow$ env.observe()
        $a \leftarrow \pi_A(s, s_0) = f(s, s_0, \theta_A)$
        **if** $a = $ STOP **or** $t_A \geq t_{\text{Max}}$ **then**
            $s^* \leftarrow s$
            env.reset()
            **break**
        env.act($a$)
    $t_B \leftarrow 0$
    **while** True **do**
        # Bob's turn
        $s \leftarrow$ env.observe()
        **if** $s = s^*$ **or** $t_A + t_B \geq t_{\text{Max}}$ **then**
            **break**
        $t_B \leftarrow t_B + 1$
        $a \leftarrow \pi_B(s, s^*) = f(s, s^*, \theta_B)$
        env.act($a$)
    $R_A \leftarrow \gamma \max(0, t_B - t_A)$
    $R_B \leftarrow -\gamma t_B$
    policy.update($R_A, \theta_A$)
    policy.update($R_B, \theta_B$)
    **return**

---

## B    HYPERPARAMETERS USED IN THE EXPERIMENTS

For the experiments with neural networks, all parameters are randomly initialized from $\mathcal{N}(0, 0.2)$. The Hyperparameters of RMSProp are set to 0.97 and $1e-6$. The other hyperparameter values used in the experiments are shown in Table 1. In some cases, we used different parameters for self-play and target task episodes. Entropy regularization is implemented as an additional cost maximizing the entropy of the softmax layer. In the StarCraft, skipping 23 frames roughly matches to one action per second.

---

[5]Experiments in VIME and SimHash papers skip 50 frames, but we matched the total number of frames in an episode by reducing the number of steps.

---

**Algorithm 2** Pseudo code for training an agent on a target task episode

**function** TARGETTASKEPISODE($t_{\text{Max}}, \theta_B$)
    $t \leftarrow 0$
    $R \leftarrow 0$
    **while** True **do**
        $t \leftarrow t + 1$
        $s \leftarrow$ env.observe()
        $a \leftarrow \pi_B(s, \emptyset) = f(s, \emptyset, \theta_B)$
        **if** env.done() **or** $t \geq t_{\text{Max}}$ **then**
            **break**
        env.act($a$)
        $R = R+$ env.reward()
    policy.update($R, \theta_B$)
    **return**

---

| Hyperparameter name | Long Hallway | Mazebase | Mountain Car | Swimmer Gather | StarCraft |
|---|---|---|---|---|---|
| Learning rate | 0.1 | 0.003 | 0.003 | 0.003 | 0.003 |
| Batch size | 16 | 256 | 128 | 256 | 32 |
| Max steps of episode ($t_{\text{max}}$) | 30 | 80 | 500 | TT: 166 SP: 200 | 200 |
| Entropy regularization | 0 | 0.003 | 0.003 | TT: 0 SP: 0.003 | TT: 0 SP: 0.003 |
| Self-play reward scale ($\gamma$) | 0.033 | 0.1 | 0.01 | 0.01 | 0.01 |
| Self-play percentage | - | 20% | 1% | 10% | 10% |
| Self-play mode | Reverse | Both | Repeat | Reverse | Repeat |
| Frame skip | 0 | 0 | 0 | 150 [5] | 23 |

Table 1: Hyperparameter values used in experiments. TT=target task, SP=self-play

## C  MAZEBASE

The agent has full visibility of the maze when the light is on. If light is off, the agent can only see the light switch. In self-play, Bob does not need to worry about things that are invisible to him. For example, if Alice started with light "off" in reverse self-play, Bob does not need to match the state of the door, because it would be invisible to him when the light is off.

In the target task, the agent and the goal are always placed on opposite sides of the wall. Also, the light and key switches are placed on the same side as the agent, but the light is always off and the door is closed initially. Therefore, in order to succeed, the agent has to turn on the light, toggle the key switch to open the door, pass through it, and reach the goal flag.

Both Alice and Bob's policies are modeled by a fully-connected neural network with two hidden layers each with 100 and 50 units (with tanh non-linearities) respectively. The encoder into each of the networks takes a bag of words over (objects, locations); that is, there is a separate word in the lookup table for each (object, location) pair. Action probabilities are output by a linear layer followed by a softmax.

### C.1  BIASING FOR OR AGAINST SELF-PLAY

The effectiveness of our approach depends in part on the similarity between the self-play and target tasks. One way to explore this in our environment is to vary the probability of the light being off initially during self-play episodes[6]. Note that the light is always off in the target task; if the light is usually on at the start of Alice's turn in reverse, for example, she will learn to turn it off, and then Bob will be biased to turn it back on. On the other hand, if the light is usually off at the start of Alice's turn in reverse, Bob is strongly biased against turning the light on, and so the test task

---

[6]The initial state of the light should dramatically change the behavior of the agent: if it is on then agent can directly proceed to the key.

becomes especially hard. Thus changing this probability gives us some way to adjust the similarity between the two tasks.

Fig. 6 (left) shows what happens when p(Light off)=0.3. Here reverse self-play works well, but repeat self-play does poorly. As discussed above, this flipping, relative to the previous experiment, can be explained as follows: low p(Light off) means that Bob's task in reverse self-play will typically involve returning the light to the on position (irrespective of how Alice left it), the same function that must be performed in the target task. The opposite situation applies for repeat self-play, where Bob needs to encounter the light typically in the off position to help him with the test task.

In Fig. 6 (right) we systematically vary p(Light off) between 0.1 and 0.9. The y-axis shows the speed-up (reduction in target task episodes) relative to training purely on the target-task for runs where the reward goes above -2. Unsuccessful runs are given a unity speed-up factor. The curves show that when the self-play task is not biased against the target task it can help significantly.

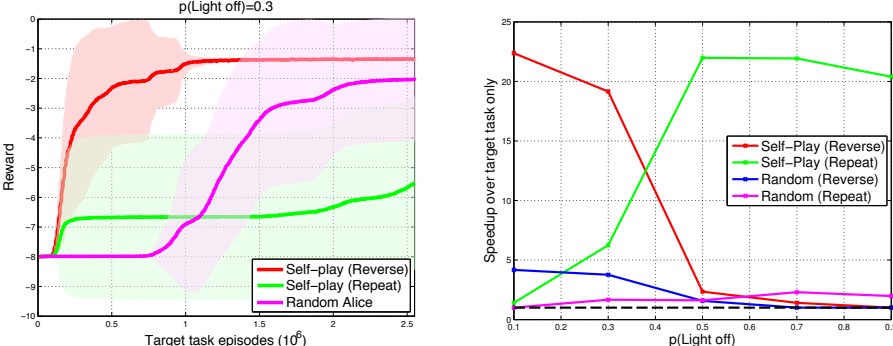

Figure 6: Left: The performance of self-play when p(Light off) set to 0.3. Here the reverse form of self-play works well (more details in the text). Right: Reduction in target task episodes relative to training purely on the target-task as the distance between self-play and the target task varies (for runs where the reward goes above -2 on the Mazebase task – unsuccessful runs are given a unity speed-up factor). The $y$ axis is the speedup, and $x$ axis is p(Light off). For reverse self-play, the low p(Light off) corresponds to having self-play and target tasks be similar to one another, while the opposite applies to repeat self-play. For both forms, significant speedups are achieved when self-play is similar to the target tasks, but the effect diminishes when self-play is biased against the target task.

## D SWIMMERGATHER EXPERIMENT

In Fig. 7 shows details of a single training run. The changes in Alice's behavior, observed in Fig. 7(c) and (d), correlate with Alice and Bob's reward (Fig. 7(b)) and, initially at least, to the reward on the test target (Fig. 7(a)). In Fig. 8 we visualize for a single training run the locations where Alice hands over to Bob at different stages of training, showing how the distribution varies.

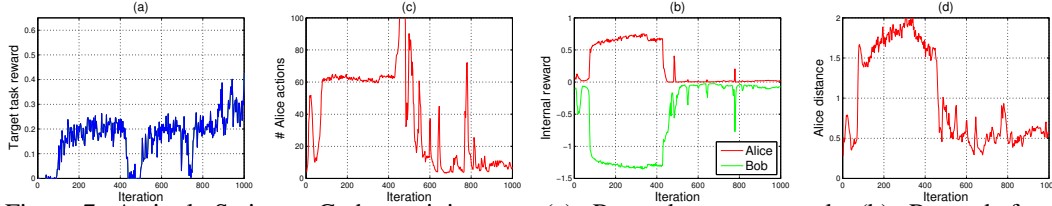

Figure 7: A single SwimmerGather training run. (a): Rewards on target task. (b): Rewards from reversible self-play. (c): The number of actions taken by Alice. (d): Distance that Alice travels before switching to Bob.

In the TRPO experiment, we used step size 0.01 and damping coefficient 0.1. The batch consists of 50,000 steps, of which 25% comes from target task episodes, while the remaining 75% is from

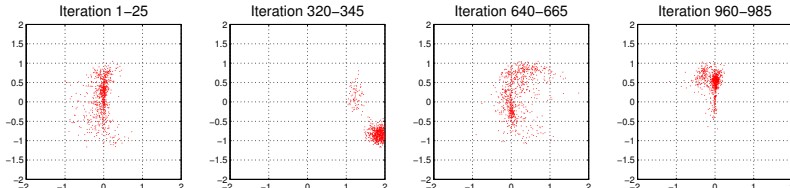

Figure 8: Plot of Alice's location at time of STOP action for the SwimmerGather training run shown in Fig. 7, for different stages of training. Note how Alice's distribution changes as Bob learns to solve her tasks.

self-play. The self-play reward scale $\gamma$ set to 0.005. We used two separate network for actor and critic, and the critic network has L2 weight regularization with coefficient of $1e-5$.

## E    STARCRAFT EXPERIMENT

We call units that perform action as active unit. This includes worker units (SCVs), the command center, and barrack. The agent controls multiple active units in parallel. At each time step, an action of $i$'th unit is output by

$$a_t^i = \pi(s_t^i, \hat{s}_t),$$

where $s_t^i$ is a unit specific local observation, and $\hat{s}_t$ is an global observation. With $s_t^i$, a unit can see the 64x64 area around it with a resolution of 4 (unit's type is also visible). The global observation contains the number of units and accumulated minerals in the game

$$\hat{s}_t = \{\lfloor N_{\text{ore}}/25 \rfloor, N_{\text{SCV}}, N_{\text{Barrack}}, N_{\text{SupplyDepot}}, N_{\text{Marines}}\}.$$

In self-play, Bob perceives only the global observation of his target state

$$\pi_B(s_t^i, \hat{s}_t, \hat{s}^*),$$

where $\hat{s}^*$ is the final global observation of Alice. Bob will succeed only if

$$\forall i \quad \hat{s}_t[i] \geq \hat{s}^*[i].$$

Table 2 shows the action space of different unit types controlled by the agent. The number of possible action is the same for all units since they controlled by a single model (unit type is encoded in the observation), but the meaning of actions differ according to unit type. An empty cell mean that the unit does nothing (nothing is sent to StarCraft, so the previous action persists).

The more complexes actions "mine minerals", "build a barracks", "build a supply depot" have the following semantics, respectively: mine the mineral closest to the current unit, build a barracks at the position of the current unit, build a supply depot on the position of the current unit.

Some actions are ignored under certain conditions: "mining" action is ignored if the distance to the closest mineral is greater than 12; "switch to Bob" is ignored if Bob is already in control; "building" and "training" actions are ignored if there is not enough resources; the actions that create a new SCV or a barracks are ignored if the number of active units is reached the limit of 10. Also "build" actions will be ignored if there is not enough room to build at the unit's location.

For the count-based exploration, we gave an extra reward of $\alpha/\sqrt{N(\hat{s}_t)}$ at every step, where $N$ is the visit count function and $\hat{s}_t$ is a global observation. We found $\alpha = 0.1$ to be works the best.

In Fig. 9 we show the result of an additional experiment where we extended the length of the episode from 200 to 300, giving more time to the agent for development. The self-play still outperforms baselines methods. Note that to make more than 6 marines, an agent has to build a supply depot as well as a barracks.

| Action ID | SCV | Command center | Barraks |
|---|---|---|---|
| 1 | move to right | train SCV | train a marine |
| 2 | move to left | switch to Bob | |
| 3 | move to top | | |
| 4 | move to bottom | | |
| 5 | mine minerals | | |
| 6 | build a barracks | | |
| 7 | build a supply depot | | |

Table 2: Action space of different unit types in StarCraft.

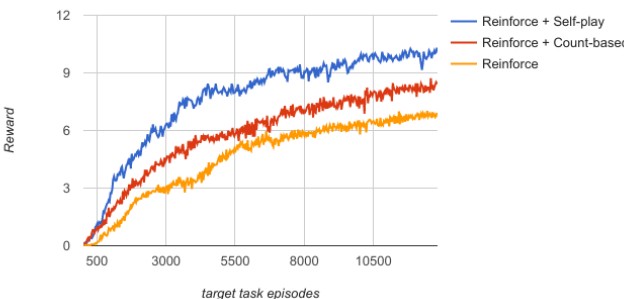

Figure 9: Plot of reward on the StarCraft sub-task of training where episode length $t_{\mathrm{Max}}$ is increased to 300.

## F FURTHER DISCUSSION

### F.1 META-EXPLORATION FOR ALICE

We want Alice and Bob to explore the state (or state-action) space, and we would like Bob to be exposed to many different tasks. Because of the form of the standard reinforcement learning objective (expectation over rewards), Alice only wants to find the single hardest thing for Bob, and is not interested in the space of things that are hard for Bob. In the fully tabular setting, with fully reversible dynamics or with resetting, and without the constraints of realistic optimization strategies, we saw in section 2.2 that this ends up forcing Bob and Alice to learn to make any state transition as efficiently as possible. However, with more realistic optimization methods or environments, and with function approximation, Bob and Alice can get stuck in sub-optimal minima.

For example, let us follow the argument in the third paragraph of Sec. 2.2, and assume that Bob and Alice are at an equilibrium (and that we are in the tabular, finite, Markovian setting), but now we can only update Bob's and Alice's policy locally. By this we mean that in our search for a better policy for Bob or Alice, we can only make small perturbations, as in policy gradient algorithms. In this case, we can only guarantee that Bob runs a fast policy on challenges that Alice has non-zero probability of giving; but there is no guarantee that Alice will cover all possible challenges. With function approximation instead of tabular policies, we cannot make any guarantees at all.

Another example with a similar outcome but different mechanism can occur using the reverse game in an environment without fully reversible dynamics. In that case, it could be that the shortest expected number of steps to complete a challenge $(s_0, s_T)$ is longer than the reverse, and indeed, so much longer that Alice should concentrate all her energy on this challenge to maximize her rewards. Thus there could be equilibria with Bob matching the fast policy only for a subset of challenges even if we allow non-local optimization.

The result is that Alice can end up in a policy that is not ideal for our purposes. In figure 8 we show the distributions of where Alice cedes control to Bob in the swimmer task. We can see that Alice has

a preferred direction. Ideally, in this environment, Alice would be teaching Bob how to get from any state to any other efficiently; but instead, she is mostly teaching him how to move in one direction.

One possible approach to correcting this is to have multiple Alices, regularized so that they do not implement the same policy. More generally, we can investigate objectives for Alice that encourage her to cover a wider distribution of behaviors.

### F.2 COMMUNICATING VIA ACTIONS

In this work we have limited Alice to propose tasks for Bob by doing them. This limitation is practical and effective in restricted environments that allow resetting or are (nearly) reversible. It allows a solution to three of the key difficulties of implementing the basic idea of "Alice proposes tasks, Bob does them": parameterizing the sampling of tasks, representing and communicating the tasks, and ensuring the appropriate level of difficulty of the tasks. Each of these is interesting in more general contexts. In this work, the tasks have incentivized efficient transitions. One can imagine other reward functions and task representations that incentivize discovering statistics of the states and state-transitions, for example models of their causality or temporal ordering, cluster structure.

