# OpenReview forum: "Intrinsic Motivation and Automatic Curricula via Asymmetric Self-Play"
_ICLR.cc/2018/Conference — Accept (Poster)_

### Official Review · AnonReviewer1 · 2017-11-27
**Good work.**

**Rating:** 8
**Confidence:** 4

**Review:**

In this paper, the authors describe a new formulation for exploring the environment in an unsupervised way to aid a specific task later. Using two “minds”, Alice and Bob, where the former proposes increasingly difficult tasks and the latter tries to accomplish them as fast as possible, the learning agent Bob can later perform a given task faster having effectively learned the environment dynamics from playing the game with Alice.

The idea of unsupervised exploration has been visited before. However, the paper presents a novel way to frame the problem, and shows promising results on several tasks. The ideas are well-presented and further expounded in a systematic way. Furthermore, the crux of the proposal and simple and elegant yet leading to some very interesting results. My only complaint is that some of the finer implementation details seems to have been omitted. For example, the parameter update equation is section 4 is somewhat opaque and requires more discussion than the motivation presented in the preceding paragraph.

Typos and grammatical errors: let assume (section 2.2), it is possible show (section 5).

Overall, I think the paper presents a novel and unique idea that would be interesting to the wider research community.

---

### Official Review · AnonReviewer2 · 2017-11-29
**baseline**

**Rating:** 5
**Confidence:** 3

**Review:**

This paper proposes an interesting model of self-play where one agent learns to propose tasks that are easy for her but difficult for an opponent. This creates a moving target of self-play objectives and learning curriculum.

The idea is certainly elegant and clearly described.
I don't really feel qualified to comment on the novelty, since this paper is somewhat out of my area of expertise, but I did notice that the authors' own description of Baranes and Oudeyer (2013) seems very close to the proposal in this paper. Given the existence of similar forms of self-play the key issue with paper I see is that there is no strong self-play baseline in the experimental evaluation. It is hard to tell whether this neat idea is really an improvement.

Is progress guaranteed? Is it not possible for Alice to imemdiately find an easy task for her where Bob times out, gets no reward signal, and therefore is unable to learn anything? Then repeating that task will loop forever without progress. This suggests that the adversarial setting is quite brittle.

I also find that the paper is a little light on the technical side.

---

> ### Author Response · Authors · 2018-01-03
> **clarification**
>
> We thank the reviewer for the constructive review. However, we would like to address several points raised:
>
> “Baranes and Oudeyer (2013) seems very close to the proposal in this paper. Given the existence of similar forms of self-play the key issue with paper I see is that there is no strong self-play baseline in the experimental evaluation”.
> - In B & O, one needs to construct a set of all possible tasks in the environment, and parameterize this set in such a way that it can be reasonably partitioned and sampled.  It is not obvious how to do this in our problems without using extra domain knowledge. In our approach, however, tasks are discovered by an agent acting in the environment, thus eliminating the need of domain knowledge about the environment. For this reason, we cannot directly compare to the approach of B & O and no other forms of self-play exist, as far as we are aware.
> Also note that it is not clear how to obtain the same sorts of guarantees we get in the tabular setting (and the related intuitions about what the learning protocol is achieving) with their method.
>
> “Is it not possible for Alice to immediately find an easy task for her where Bob times out, gets no reward signal, and therefore is unable to learn anything? Then repeating that task will loop forever without progress. This suggests that the adversarial setting is quite brittle”.
> - An easy task means it only requires few actions for Alice to succeed. In repeat self-play, this means that Bob would only need a few actions to succeed also. So it is unlikely that Bob will keep failing on such easy tasks since even taking random actions would sometimes yield success on such easy tasks. The same is true for the reverse self-play because of the reversibility assumption (Bob just needs to perform the opposite of Alice's actions in reverse order).
> In general however, our adversarial setting does assume that Alice and Bob are trained in sync. Similar to generative adversarial networks, if one of them gets too far ahead of the other, then it can impede training. However, our experiments demonstrate that the two of them can be successfully training is possible on non-trivial problems.
>
> “I also find that the paper is a little light on the technical side”.
> -We will add further technical details in the final revision.

---

### Official Review · AnonReviewer3 · 2017-11-29
**Compelling approach with solid results on a reasonable set of baselines**

**Rating:** 8
**Confidence:** 4

**Review:**

The paper presents a method for learning a curriculum for reinforcement learning tasks.The approach revolves around splitting the personality of the agent into two parts. The first personality learns to generate goals for other personality for which the second agent is just barely capable--much in the same way a teacher always pushes just past the frontier of a student’s ability. The second personality attempts to achieve the objectives set by the first as well as achieve the original RL task.

The novelty of the proposed method is introduction of a teacher that learns to generate a curriculum for the agent.The formulation is simple and elegant as the teacher is incentivised to widen the gap between bob but pays a price for the time it takes which balances the adversarial behavior.

Prior and concurrent work on learning curriculum and intrinsic motivation in RL rely on GANs (e.g., automatic goal generation by Held et al.), adversarial agents (e.g., RARL by Pinto et al.), or algorithmic/heuristic methods (e.g., reverse curriculum by Florensa et al. and HER Andrychowicz et al.).  In the context of this work, the contribution is the insight that an agent can be learned to explore the immediate reachable space but that is just within the capabilities of the agent. HER and goal generation share the core insight on training to reach goals. However, HER does generate goals beyond the reachable it instead relies on training on existing reached states or explicitly consider the capabilities of the agent on reaching a goal. Goal generation while learning to sample from the achievable frontier does not ensure the goal is reachable and may not be as stable to train.

As noted by the authors the above mentioned prior work is closely related to the proposed approach. However, the paper only briefly mentions this corpus of work. A more thorough comparison with these techniques should be provided even if somewhat concurrent with the proposed method. The authors should consider additional experiments on the same domains of this prior work to contrast performance.

Questions:
Do the plots track the combined iterations that both Alice and Bob are in control of the environment or just for Bob?

---

> ### Author Response · Authors · 2018-01-05
> **answer to the question**
>
> “Do the plots track the combined iterations that both Alice and Bob are in control of the environment or just for Bob?”
> - The plots track the iterations/steps of Bob during target task episodes where a supervision from the environment given as a reward signal. The paradigm we consider is the RL equivalent of semi-supervised learning, with self-play being the unsupervised learning component. In this context, what matters is the number of labeled examples (analogously: target task episodes) used, rather than the number of unlabeled points (i.e. self-play episodes). This RL paradigm was introduced in Finn et al. 2016 https://arxiv.org/abs/1612.00429 and we note that they also use this convention. We will clarify this in the final version.

---

### Decision · Program_Chairs · 2018-01-29
**ICLR 2018 Conference Acceptance Decision**

**Decision:**

Accept (Poster)

**Comment:**

I fully agree with strong positive statements in the reviews.  All reviewers agree that the paper introduces a novel and elegant twist on standard RL, wherein one agent proposes a sequence of diverse tasks to a second agent so as to accelerate the second agent's learning models of the environment.  I also concur that the empirical testing of this method is quite good.  There are strong and/or promising results in five different domains (Hallway, LightKey, MountainCar, Swimmer Gather and TrainingMarines in StartCraft). This paper would make for a strong poster at ICLR.